# Treatment of Metaphyseal Defects in Plated Proximal Humerus Fractures with a New Augmentation Technique—A Biomechanical Cadaveric Study

**DOI:** 10.3390/medicina59091604

**Published:** 2023-09-05

**Authors:** Daniel Zhelev, Stoyan Hristov, Ivan Zderic, Stoyan Ivanov, Luke Visscher, Asen Baltov, Simeon Ribagin, Karl Stoffel, Franz Kralinger, Jörg Winkler, R. Geoff Richards, Peter Varga, Boyko Gueorguiev

**Affiliations:** 1AO Research Institute Davos, 7270 Davos, Switzerland; zhelev_dr@mail.bg (D.Z.); ivan.zderic@aofoundation.org (I.Z.); luke.visscher@uqconnect.edu.au (L.V.); geoff.richards@aofoundation.org (R.G.R.); peter.varga@aofoundation.org (P.V.); 2Department of Orthopedics and Traumatology, University Hospital for Active Treatment, 8018 Burgas, Bulgaria; hristov@dr.com; 3Department of Orthopaedics and Traumatology, Medical University of Varna, 9002 Varna, Bulgaria; ton_ivanov@abv.bg; 4School of Medicine, Queensland University of Technology, Brisbane 4000, Australia; 5Department of Trauma Surgery, University Multiprofile Hospital for Active Treatment and Emergency Medicine ‘N. I. Pirogov’, 1606 Sofia, Bulgaria; asen_b@bitex.bg; 6Department of Health Pharmaceutical Care, Medical College, University ‘Prof. Dr. Asen Zlatarov’, 8010 Burgas, Bulgaria; simribagin@gmail.com; 7Department of Orthopaedics and Traumatology, University Hospital Basel, 4031 Basel, Switzerland; karl.stoffel@usb.ch; 8Department of Trauma Surgery, Medical University of Vienna, 1090 Vienna, Austria; franz.kralinger@wienkav.at; 9Trauma and Sports Department, Ottakring Clinic, Teaching Hospital, Medical University of Vienna, 1160 Vienna, Austria; 10Cantonal Hospital Graubuenden, 7000 Chur, Switzerland; jwinkler@spitaldavos.ch

**Keywords:** bone cement augmentation, metaphyseal void defect, novel technique, orthopaedic plating, proximal humerus fracture

## Abstract

*Background and Objectives*: Unstable proximal humerus fractures (PHFs) with metaphyseal defects—weakening the osteosynthesis construct—are challenging to treat. A new augmentation technique of plated complex PHFs with metaphyseal defects was recently introduced in the clinical practice. This biomechanical study aimed to analyze the stability of plated unstable PHFs augmented via implementation of this technique versus no augmentation. *Materials and Methods*: Three-part AO/OTA 11-B1.1 unstable PHFs with metaphyseal defects were created in sixteen paired human cadaveric humeri (average donor age 76 years, range 66–92 years), pairwise assigned to two groups for locked plate fixation with identical implant configuration. In one of the groups, six-milliliter polymethylmethacrylate bone cement with medium viscosity (seven minutes after mixing) was placed manually through the lateral window in the defect of the humerus head after its anatomical reduction to the shaft and prior to the anatomical reduction of the greater tuberosity fragment. All specimens were tested biomechanically in a 25° adduction, applying progressively increasing cyclic loading at 2 Hz until failure. Interfragmentary movements were monitored by motion tracking and X-ray imaging. *Results*: Initial stiffness was not significantly different between the groups, *p* = 0.467. Varus deformation of the humerus head fragment, fracture displacement at the medial humerus head aspect, and proximal screw migration and cut-out were significantly smaller in the augmented group after 2000, 4000, 6000, 8000 and 10,000 cycles, *p* ≤ 0.019. Cycles to 5° varus deformation of the humerus head fragment—set as a clinically relevant failure criterion—and failure load were significantly higher in the augmented group, *p* = 0.018. *Conclusions*: From a biomechanical standpoint, augmentation with polymethylmethacrylate bone cement placed in the metaphyseal humerus head defect of plated unstable PHFs considerably enhances fixation stability and can reduce the risk of postoperative complications.

## 1. Introduction

Proximal humerus fractures (PHFs) are with second most frequent incidence related to upper extremity, following distal forearm fractures. They represent approximately 5% of all fractures, display constantly increasing incidence [1], occur mostly in the population of over 65 years of age as a result of falls, and have a higher prevalence in females versus males [2]. Epidemiologic changes with increasing severity of osteoporosis result in more common displaced PHFs. Approximately 20% of all PHFs are unstable and displaced [3], benefiting from the advantages of operative treatment. Special attention is required by three- and four-part fractures according to the Neer Classification [4]. Different methods for open reduction and internal fixation (ORIF) exist, with locked plating being the standard treatment of choice for osteoporotic and/or comminuted PHFs due to the angular stable construct without plate-to-bone compression, enhancing vascularization and periosteal blood supply [5,6,7]. However, despite the biomechanical advantages, the postoperative complication rate is still high [8,9,10]. Among others, the most common complications are represented by varus collapse, screw cut-through and screw penetration [11,12], frequently indicated in cases with medial comminution or in elderly patients with poor bone stock [13]. A prerequisite for such complications is the reduced mechanical support due to fracture morphology [14,15]. On the other hand, the high rigidity of the locking plates could be another reason for cutting of the screws through the osteoporotic bone and subsidence of the humerus head fragment while the screws remain locked in position.

Both screw cut-through and perforation risk implant penetration into the joint cavity, resulting in damage to the articular strictures, impaired joint function, and persistent pain. The penetration of the screws into the articulating joint of the humerus head carries the risk of glenoid wear and erosion [13,16].

Studies have demonstrated a direct relationship of the medial calcar communication and local vascularization with the development of subsequent avascular necrosis (AVN) [17]. According to a series of clinical and biomechanical reports, in case of poor bone quality, the challenging stable fixation can be considerably enhanced via placement of an allograft or an autograft [18,19].

In elderly patients, a posttraumatic humerus head defect, resulting from bone loss and located in the metaphyseal region, frequently exists and could be filled in with supplemental materials such as autograft, allograft or bone cement [20]. Of all materials used for defect filling in the clinical practice, the highest biomechanical stability is achieved with polymethylmethacrylate (PMMA)-based bone cements [21], in contrast to calcium phosphate and calcium sulphate cements that are biologically advantageous since they are able to promote bone healing and address problems of vascularization but capable of providing only limited biomechanical support, leading to increased risk of early fixation failure in elderly patients [22]. The filling of the bone defect with the cement causes the screws crossing it to become loaded along their entire length, thus distributing the bone strain more evenly, and additionally providing a better implant anchorage to the bone via an increase in the bone–implant contact interface.

A new augmentation technique of plated complex PHFs with metaphyseal defects was recently introduced in clinical practice (Figure 1) [23]. Nevertheless, the biomechanical competence of augmented plate fixation with this technique versus non-augmented plating has not been investigated so far.

Therefore, this biomechanical cadaveric study aimed to analyze the stability of plated unstable PHFs augmented via implementation of this technique versus no augmentation. It was hypothesized that the augmentation of the humerus head defect would provide superior fixation stability and reduce varus deformation as well as screw cut-through and perforation.

## 2. Materials and Methods

### 2.1. Specimens and Preparation

Eight pairs of fresh-frozen (−20 °C) human cadaveric humeri from 4 female and 4 male donors aged 76 ± 8 years (mean ± standard deviation, SD) (range 66–92 years) were used. This study was approved by the institutional internal review board based on the approval of the specimen delivery by Science Care Ethics Committee. All donors provided confirmation of their informed consent inherent within the donation of the anatomical gift statement during their lifetime. The specimens were stripped of all soft tissues and underwent computed tomography (CT) scanning (Revolution EVO, GE Healthcare, Chicago, IL, USA) at 0.63 mm slice thickness. Volumetric bone mineral density (BMD) was calculated in the cancellous bone of the humerus head using a phantom (European Forearm Phantom QRM-BDC/6, QRM GmbH, Möhrendorf, Germany).

A three-part PHF according to the Neer Classification (AO/OTA 11-B1.1) was created in all specimens by means of three osteotomies.

Based on clinical evidence, this fracture type, representing severe bone damage with communication of the inferomedial cortex, is predominant for patients within the above-mentioned age range [24]. The first osteotomy was set parallel to the surgical neck at a distance of 8 mm below the epiphysis. The second one was performed to reproduce a superomedial wedge opening at an angle of 15° to the anteroposterior pivot axis located at the most lateral aspect of the first osteotomy. The third osteotomy was set along the border of the greater tuberosity–intertubercular groove (Sulcus Bicipitalis) with separation of the greater tuberosity (GT) from both the humerus head and shaft. A natural metaphyseal defect with distinct borders filled with a substance of reduced bone quality was identified in all specimens through the traumatic lateral window, the latter resulting from the simulation of the fracture pattern via the third osteotomy.

Within each of the eight pairs of specimens, one humerus was assigned for locked plating without further interventions, whereas the contralateral part was assigned for locked plating with supplemental bone cement augmentation of the metaphyseal defect, resulting in two respective treatment groups—control and augmented—consisting of eight specimens each (n = 8), with an equal number of right and left sides and identical screw configuration. A minimum sample size of 8 specimens per group was identified in a priori power analysis with a statistical power of 0.8 at a level of significance of 0.05 under the assumption that the SD in each group was not larger than 100% of the difference between the mean values of the groups.

In the augmented group, six-milliliter cement with medium viscosity (TRAUMACEM^TM^ V+, DePuy Synthes, Raynham, MA, USA; chemical composition: 45.0% PMMA, 40.0% zirconium dioxide, 14.5% hydroxyapatite, 0.5% benzoyl peroxide [25]; seven minutes after mixing) was placed manually through the lateral window in the defect of the humerus head fragment after its anatomical reduction to the shaft and prior to the anatomical reduction of the GT fragment in a manner resembling clinical practice [23].

In both study groups, the temporary fixation of the reduced fragments was performed using pointed forceps and 1.6 mm Kirschner wires.

The locked plating of each specimen was performed using a short PHILOS plate (DePuy Synthes, Zuchwil, Switzerland) positioned and attached to the humerus with reposition clamps according to the surgical technical guide, and fixed to it with a frequently used screw configuration including six proximal locking screws (plate rows A, C, and E, Figure 2) and three distal locking screws based on previous studies [15]. After the drilling of all pilot holes, plate rows E and C were first occupied, followed by insertion of two screws into row A and three distal screws. All screws were tightened at 1.5 Nm using a torque limiter. In the augmented group, the screws in rows C and E were inserted after hardening of the bone cement and—in contrast to row A—they passed through it in all humeri (Figure 2).

Next, the humerus shafts of all specimens were cut distally at a 175 mm distance from the top of the humerus head, and the distal 60 mm of each humerus was embedded in a PMMA (SCS-Beracryl D28, Swiss Composite, Jegenstorf, Switzerland) cylindrical block. Finally, optical marker sets were attached to the humerus head, GT and shaft fragments of each specimen for motion tracking.

### 2.2. Biomechanical Testing

Biomechanical testing was performed at room temperature on a material testing machine (MiniBionix, MTS Systems Corp., Eden Prairie, MN, USA) equipped with a 4 kN load cell (HUPPERT 6, HUPPERT GmbH, Herrenberg, Germany). The setup was adopted from previously published work [26] (Figure 3). The humerus shaft of each specimen was aligned at a 25° adduction and connected to the machine base via a custom fixation clamp firmly holding the PMMA embedding. A concave PMMA shell, interconnected to the machine transducer via an XY table, was used for axial force transmission to the humerus head. The XY table was implemented to balance all shear forces and bending moments.

The loading protocol commenced with a non-destructive quasi-static ramp from 20 N preload to 200 N compression at 18 N/s. Afterwards, progressively increasing cyclic axial loading with a physiological loading profile of each cycle was applied at 2 Hz [27]. Whereas the valley load of each cycle was held constant at 50 N throughout testing, the peak load, starting at 200 N, was increased at a rate of 0.05 N/cycle [26] until a 15 mm displacement of the machine actuator relative to the test start. This stop criterion proved to be relevant for destructive testing with catastrophic specimen’s failure [26].

### 2.3. Data Evaluation and Analysis

Axial load and axial displacement were collected throughout testing at 200 Hz. Construct stiffness was calculated as the slope of the load–displacement curve derived from the initial quasi-static ramp within a linear range of 100–180 N.

The three-dimensional coordinates of the optical markers were collected at 20 Hz (Aramis, Carl Zeiss GOM Metrology GmbH, Braunschweig, Germany) to evaluate the relative movements of the humerus head to the shaft fragment, and of the GT to the humerus head fragment.

The following parameters of interest were analyzed after 2000, 4000, 6000, 8000 and 10,000 test cycles under valley loading with respect to the start of the cyclic test using motion tracking data: (1) varus deformation, i.e., the magnitude of the angular head-to-shaft movement around the anteroposterior axis; (2) head displacement, i.e., the magnitude of the fracture displacement at the most medial aspect of the humerus head fragment relative to the shaft; (3) GT displacement, i.e., the fracture displacement magnitude at the most proximal aspect of the GT fragment relative to the humerus head fragment; and (4) head rotation, i.e., the magnitude of the angular humerus head fragment rotation around the humerus head axis. As a note, 10,000 was the highest rounded number of cycles with no indicated failure of any specimen. A varus deformation of 5° under valley loading was considered to set a clinically relevant criterion for failure [28]. The number of cycles until its fulfillment under valley loading—i.e., cycles to failure—was evaluated along with the corresponding peak load—i.e., failure load.

In addition, anteroposterior X-ray images were taken using a triggered C-arm (Siemens Cios Select; Siemens AG, Erlangen, Germany) at timed intervals of 500 cycles during the cyclic test under valley loading. Based on the radiological data, the following parameters of interest were analyzed after 2000, 4000, 6000, 8000 and 10,000 test cycles under valley loading with respect to the start of the cyclic test using a Matlab software package (V.2022, MathWorks, Natick, MA, USA): (1) screw migration plate rows A and C, i.e., the change in distance between the screw tip and the intersection point of the projected line along the screw axis with the outer border of the humerus head; (2) screw cut-out rows A and C, i.e., the change in distance between the screw tip and the intersection point of the projected line which passes through the screw tip and is oriented perpendicular to the screw axis in the frontal plain, with the outer border of the humerus head; and (3) screw bending rows A and C, i.e., the angular deviation of the screw axis in the frontal plane relative to the plate resulting from the connection of the screw head to plate.

Statistical analysis was performed with an SPSS software package (V.27, IBM SPSS Statistics, Armonk, NY, USA). Normality of data distribution was screened and proved with the Shapiro–Wilk test. Construct stiffness, cycles to failure and failure load were compared between the groups with the Paired-Samples *t*-test. Varus deformation, head and GT displacements, head rotation, screw migration, screw cut-out, and screw bending were compared between the groups and statistically evaluated over the course of cycles via the Paired-Samples *t*-test and the General Linear Model Repeated Measures test, respectively. The level of significance was set to 0.05.

## 3. Results

BMD (mgHA/cm^3^) was 148.7 ± 31.3 in the control and 154.2 ± 34.7 in the augmented group, with no significant differences between them (*p* = 0.896).

Similarly, construct stiffness (N/mm) was 423.5 ± 121.0 in the control and 445.1 ± 132.1 in the augmented group, without significant differences between them (*p* = 0.467).

The outcome measures of the parameters evaluated over the five time points after 2000, 4000, 6000, 8000 and 10,000 cycles are presented in Table 1.

Varus deformation, head displacement and GT displacement increased significantly between 2000 and 10,000 cycles in each group (*p* ≤ 0.024) and were significantly smaller in the augmented versus control group (*p* ≤ 0.004). In contrast, head rotation increased significantly between 2000 and 10,000 cycles in each group (*p* ≤ 0.018) but remained non-significantly different between the groups (*p* = 0.314).

Screw migration and screw cut-out rows A and C increased significantly between 2000 and 10,000 cycles in the control group (*p* ≤ 0.034) but not in the augmented group (*p* ≥ 0.110) and were significantly smaller in the augmented versus the control group (*p* ≤ 0.019).

Screw bending row A increased significantly between 2000 and 10,000 cycles in each group (*p* ≤ 0.013) and remained non-significantly different between the groups (*p* = 0.361).

Screw bending row C increased significantly between 2000 and 10,000 cycles in each group (*p* ≤ 0.025) and was significantly smaller in the augmented versus the control group (*p* = 0.001).

Cycles to failure and failure load were 13219 ± 2642 and 860.0 ± 332.1 N in the control and 17,552 ± 2638 and 1077.6 ± 331.9 N in the augmented group, with significant difference between them (*p* = 0.018) (Figure 4).

The catastrophic failure mode of the specimens was predominantly expressed by excessive caudal humerus head displacement along the humerus shaft with an associated concomitant screw tip cut-through within the humerus head. Loosening of the screw head from the plate locking mechanism with backing-out was observed in three specimens and screw breakage was detected in two specimens. In five specimens, the proximal portion of the plate excessively bent in the varus.

## 4. Discussion

The main goals of the treatment of osteoporotic humerus fractures are achievement of sufficient implant purchase to bone allowing for early patient mobilization and physiotherapy, and prevention of implant-related complications. Locking plates have been successfully introduced and are currently frequently used for surgical treatment of PHFs—a typical injury in elderly patients associated with osteoporosis [29,30]. However, despite improved operative techniques and implant designs, the literature still reports considerable high rates of postoperative complications [31]. The predominant reasons for them are both the lack of medial calcar support due to fracture morphology and the presence of impaired bone quality because of existing osteoporosis [15,32,33,34]. That is why the augmentation of PHFs is a possible valuable option for improvement of fixation stability [35,36]. Its effect in cases with advanced osteoporosis and reduced bone stock has been demonstrated in several clinical and biomechanical studies comparing different augmentation methods [37,38,39,40]. On the other hand, although the Hertel’s criteria are the basis for AVN prognosis, they alone do not suffice for assessment, and other characteristics of the patient such as age, sex and harmful habits need to be taken into account [17].

The findings in the present study are in agreement with those of other biomechanical cadaveric work reporting enhanced stability of fixation of unstable PHFs after augmentation with PMMA-based bone cements [41]. The augmentation of posttraumatic defects within the humerus head in this work considerably decreased the varus tilting and displacement of the plated constructs, and hence the risk of such previously observed postoperative complications as screw penetration and cut-out [9]. PMMA-based bone cement was selected for application mainly because of its superior biomechanical—and not necessarily biological—characteristics in comparison to other types of bone cements such as, e.g., those based on calcium phosphate [6]. Cement augmentation also led to a reduction in both the migration and cut-out of the most proximal screws (plate row A) which are usually subjected to the highest axial loading. In addition, it was found that the augmentation resulted in considerably higher endurance of the locking mechanism of the screws passing through the bone cement because of the more even load distribution to each of them transferred to the shaft via the compound of the bone cement. It is also worth mentioning that the augmentation via filling of the metaphyseal defect of the humerus head with bone cement not only enhances the stability of fixation but can also act as a volumetric filler increasing the resistance of the humerus head fragment toward the glenoid reaction forces, thus preventing screw perforation through the articular surface.

Several clinical studies have highlighted the importance calcar screw support and its relation to decreased postoperative complication rates following plating of PHFs [33,42,43]. According to a previous microstructural analysis, the best bone stock within the humerus head is located in its medial and dorsal regions [1]. It was concluded that screws placed inferomedially have a stronger contribution to the stability of fixation versus superomedial and superolateral insertion [44]. Whereas excessively proximal positioning of the calcar screws did not improve the biomechanical characteristics, their augmentation with bone cement resulted in increased construct stiffness and failure load of the plated constructs [45]. An optimal configuration of six proximal screws with the presence of calcar screws (plate row E) and screws passing through the humerus head defect filled with cement was selected in the present study to increase the fixation stability according to previous work [46]. The lengths of all screws were selected to ensure a tip-to-joint distance of 6 mm, in agreement with a previous study [47]. This value has been reported to provide an optimal balance between stability and reduced risk of screw penetration and cut-out.

In this study, the preparation of the specimens in terms of osteotomizing, augmentation and fixation followed the treatment procedure of plated PHFs in clinical practice. The drilling of the pilot holes for the screws passing through the cement (plate rows C and E) was performed after its hardening to avoid any leakage.

One of the characteristic PMMA features is the development of an exothermic curing reaction. A series of biomechanical studies investigated the increase in temperature at the screw tips underneath the articular surface resulting from the released heat during augmentation because the cartilage cells located there have a lower critical threshold for thermal apoptosis and necrosis [39,48,49]. Although no concerns related to the curing temperature were reported, it can be expected that the placement of PMMA material at a greater distance from the articular surface via the technique in the current study could be advantageous over screw tip augmentation. In addition, it should be taken into account that the released heat during augmentation directly depends on the amount of bone cement and ambient temperature.

The current work has some limitations inherent to those of all human cadaveric studies, incapable of entirely simulating in vivo situations with surrounding soft tissue following bone fracture. A limited number of specimens with a small sample size was investigated, resulting in restriction of the translation to generalized clinical applications. In addition, although cadaveric bones allow for a better simulation of the physiological conditions, variation in age, sex, anatomy, and demographics must be considered. Moreover, bone microarchitecture in general and specifically within the proximal humerus is highly complex and diverse for each specific individual. Pairwise comparisons in this study compensated for these variations. In addition, the used simplified biomechanical test model did not consider all muscle forces and moments acting on the humerus. Further, the present study design represented biomechanical conditions without consideration of bone healing; however, on the other hand, it reflected the worst-case scenario for testing the stability of the plated constructs under dynamic loading and allowed the detection of important significant differences between the treatment groups.

Although no specific adverse effects related to the use of PMMA-based bone cement for implant augmentation via filling of metaphyseal humerus head defects were observed in the current study, the possibility for both leakage and heat-related damage always needs to be taken into consideration. In the case of the lack of trabecular bone stock in the central humerus head region, the new technique could help a surgeon in performing a treatment that is surgically safe and effective for the patient. Moreover, it might be helpful to meet the requirement for early shoulder mobilization and prevent disabling stiffness by securing the anchorage of the screw shafts passing through the bone defect. In addition, possible cartilage perforation or screw prominence over the cartilage layer can be avoided because it is no more absolutely necessary that the screw tips reach subchondral bone. Development of new bioresorbable bone cements featuring appropriate initial biomechanical stability with suitable periods of biodegradability could improve the application of the new technique.

## 5. Conclusions

From a biomechanical standpoint, augmentation with PMMA-based bone cement placed in the metaphyseal humerus head defect of plated unstable PHFs considerably enhances fixation stability and can reduce the risk of postoperative complications.

## Figures and Tables

**Figure 1 medicina-59-01604-f001:**
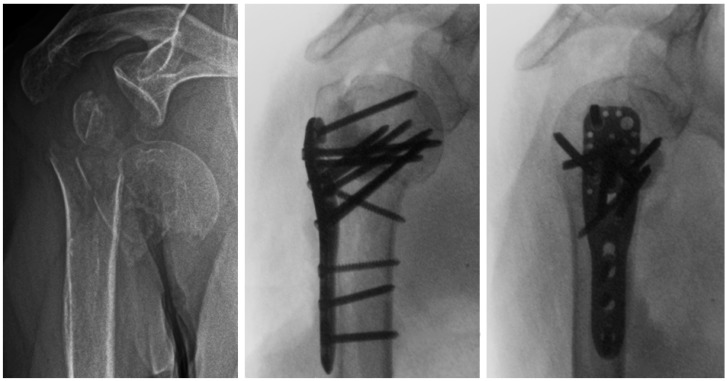
Preoperative (**left**) and intraoperative X-ray images in anteroposterior (**middle**) and lateral (**right**) views of an 87-year-old female patient with a three-part AO/OTA 11-B1.3 PHF and metaphyseal defect. Prior to plate fixation, partially cured PMMA-based bone cement with medium viscosity was placed manually in the defect through the traumatic lateral window 5 to 7 min after mixing.

**Figure 2 medicina-59-01604-f002:**
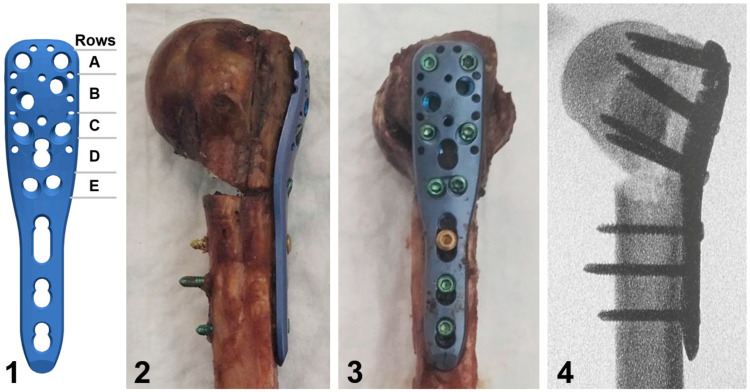
Illustration of a short PHILOS plate with labeled rows A to E of proximal screw holes (**1**), together with photographs in posterior (**2**) and lateral (**3**) views of a plated specimen from the augmented group, and the corresponding anteroposterior X-ray image (**4**) visualizing its defect filled with bone cement and screws in rows C and E passing through the latter.

**Figure 3 medicina-59-01604-f003:**
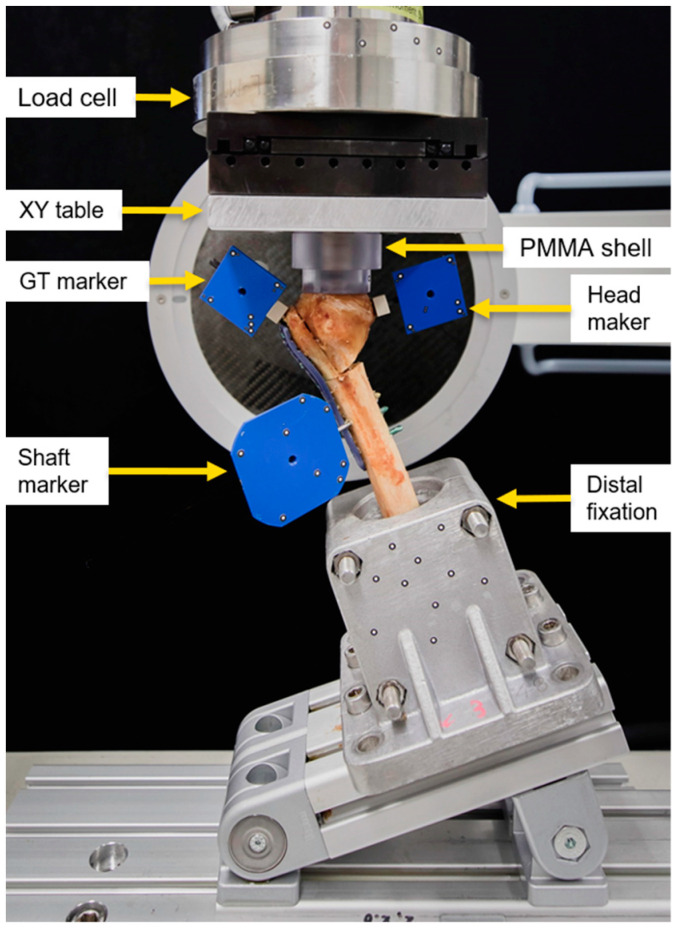
Biomechanical testing setup with a mounted specimen.

**Figure 4 medicina-59-01604-f004:**
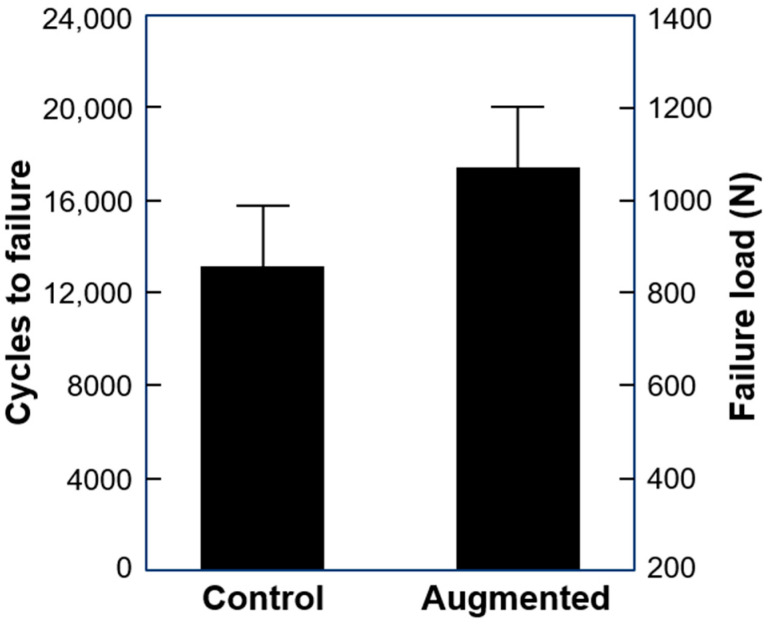
Cycles to failure and failure load in the study groups presented as mean value and SD.

**Table 1 medicina-59-01604-t001:** Outcome measures evaluated after 2000, 4000, 6000, 8000 and 10,000 cycles, presented for each group as mean value and SD, along with *p*-values from the evaluation between the groups and over cycles.

Parameter of Interest	Group	Cycles					*p*-Value Groups
		2000	4000	6000	8000	10,000	
Varus deformation (°)	Control	0.34 ± 0.28	1.14 ± 0.98	1.86 ± 1.44	2.68 ± 1.56	2.98 ± 1.62	0.004
	Augmented	0.21 ± 0.11	0.49± 0.35	0.68 ± 0.45	0.92 ± 0.56	1.34 ± 1.03	
	***p*-value cycles**	≤0.015					
Head displacement (mm)	Control	0.35 ± 0.29	1.11 ± 1.01	2.02 ± 1.29	2.88 ± 1.58	4.49 ± 1.93	0.001
	Augmented	0.26 ± 0.12	0.65 ± 0.42	1.09 ± 0.74	1.61 ± 1.04	2.37 ± 1.56	
	***p*-value cycles**	≤0.024					
GT displacement (mm)	Control	0.18 ± 0.15	0.53 ± 0.45	0.94 ± 0.81	1.36 ± 1.02	2.23 ± 1.39	0.001
	Augmented	0.11 ± 0.07	0.30 ± 0.25	0.51 ± 0.46	0.79± 0.70	1.22 ± 1.04	
	***p*-value cycles**	≤0.023					
Head rotation (°)	Control	0.23 ± 0.15	0.68 ± 0.61	1.08 ± 0.92	1.72 ± 1.09	1.94 ± 1.18	0.314
	Augmented	0.14± 0.09	0.32 ± 0.24	0.52 ± 0.41	0.79 ± 0.61	1.15 ± 0.94	
	***p*-value cycles**	≤0.018					
Screw migration row A (mm)	Control	0.82 ± 0.51	0.89 ± 0.54	1.33 ± 0.89	1.46 ± 1.15	2.63 ± 1.22	0.002
	Augmented	0.25 ± 0.14	0.46± 0.30	0.75 ± 0.64	1.02 ± 0.92	1.18 ± 1.04	
	***p*-value cycles**	0.017 (Control), 0.115 (Augmented)					
Screw migration row C (mm)	Control	0.61 ± 0.36	1.19 ± 0.75	1.34 ± 0.67	1.75 ± 0.68	2.25 ± 1.19	0.012
	Augmented	0.44 ± 0.29	0.59 ± 0.33	0.88 ± 0.45	1.01 ± 0.53	1.22 ± 0.64	
	***p*-value cycles**	0.027 (Control), 0.110 (Augmented)					
Screw cut-out row A (mm)	Control	0.69 ± 0.34	0.87 ± 0.42	1.01 ± 0.62	1.33 ± 0.97	2.33 ± 1.38	0.002
	Augmented	0.39 ± 0.28	0.55 ± 0.38	0.68 ± 0.52	0.81 ± 0.64	0.99 ± 0.76	
	***p*-value cycles**	0.034 (Control), 0.253 (Augmented)					
Screw cut-out row C (mm)	Control	0.67 ± 0.35	1.09 ± 0.44	1.26 ± 0.76	1.60 ± 1.29	3.09 ± 1.74	0.019
	Augmented	0.59 ± 0.32	0.94 ± 0.37	1.15 ± 0.64	1.22 ± 0.81	1.57 ± 0.97	
	***p*-value cycles**	0.021 (Control), 0.117 (Augmented)					
Screw bending row A (°)	Control	1.12 ± 0.58	1.20 ± 0.74	1.43 ± 0.89	2.26 ± 1.42	3.26 ± 1.75	0.361
	Augmented	0.69 ± 0.28	1.13 ± 0.58	1.34 ± 0.69	1.92 ± 0.98	2.95 ± 1.37	
	***p*-value cycles**	≤0.013					
Screw bending row C (°)	Control	1.22 ± 0.38	1.56 ± 0.41	2.18 ± 0.86	2.39 ± 0.92	2.47 ± 0.98	0.001
	Augmented	0.24 ± 0.19	0.41 ± 0.28	0.57 ± 0.31	0.88 ± 0.65	1.04 ± 0.77	
	***p*-value cycles**	≤0.025					

## Data Availability

The datasets analyzed during the current study are available from the corresponding author on reasonable request.

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
