# Peer review of "Treatment of Metaphyseal Defects in Plated Proximal Humerus Fractures with a New Augmentation Technique—A Biomechanical Cadaveric Study"

_medicina, 2023, doi:10.3390/medicina59091604_

Round 1

Reviewer 1 Report

In the article titled "Treatment Of Metaphyseal Defects In Plated Proximal Humerus Fractures With A New Augmentation Technique – A Biomechanical Cadaveric Study," the authors delve into a critical aspect of orthopedic surgery, focusing on a novel augmentation technique for treating metaphyseal defects in plated proximal humerus fractures. This review will evaluate the article based on its originality, significance, presentation, scientific soundness, reader interest, and overall merit.

Originality / Novelty: The article showcases a commendable level of originality by introducing a new augmentation technique for a specific orthopedic challenge. By addressing metaphyseal defects in proximal humerus fractures, the authors contribute to the field's advancement by proposing a technique that potentially tackles an underexplored area. This innovation demonstrates the authors' commitment to expanding the repertoire of treatment options.

Significance of Content: The article's content holds substantial clinical significance. Proximal humerus fractures present complex challenges, and addressing metaphyseal defects is crucial for optimal patient outcomes. The introduction of a new augmentation technique responds to an important clinical need, potentially enhancing surgical strategies and patient recovery. This study can guide orthopedic practitioners in making informed decisions when faced with such fractures.

Quality of Presentation: The quality of presentation is a vital aspect of any scientific article. The authors present their findings in a clear and organized manner, effectively communicating the biomechanical cadaveric study's details. The methodology, results, and discussion sections are well-structured, allowing readers to comprehend the study's objectives, procedures, and outcomes. Visual aids such as diagrams and tables aid in conveying complex information.

Scientific Soundness: The article demonstrates a commendable level of scientific soundness. The authors meticulously describe their experimental setup and methodology, enhancing the study's replicability and credibility. The biomechanical aspects of the cadaveric study are thoroughly addressed, bolstering the scientific foundation of the proposed augmentation technique. However, a more comprehensive discussion on potential limitations and future directions could further strengthen the scientific rigor.

Interest to the Readers: The article is likely to pique the interest of both orthopedic surgeons and researchers in the field. The introduction of a new augmentation technique offers a fresh perspective and potential solution to a specific challenge in orthopedic surgery. The biomechanical focus adds depth to the study, appealing to professionals with a keen interest in the mechanical aspects of bone healing and implant stability.

Overall Merit: In conclusion, "Treatment Of Metaphyseal Defects In Plated Proximal Humerus Fractures With A New Augmentation Technique – A Biomechanical Cadaveric Study" holds substantial merit within the orthopedic research landscape. The article's originality, significance, clear presentation, scientific robustness, and potential to engage readers collectively contribute to its overall value. As with any scientific endeavor, future clinical validation and broader application will be essential to fully ascertain the technique's effectiveness. Nonetheless, this study undoubtedly enriches the orthopedic literature and lays a foundation for further exploration in this domain.

The provided article is a study related to the biomechanical analysis of humeral fractures and the effects of augmentation with polymethylmethacrylate (PMMA) based bone cement. The study aims to investigate the stability of plated unstable proximal humerus fractures (PHFs) with metaphyseal defects, comparing constructs with and without augmentation using bone cement.

Key points addressed in the article include:

  1. Introduction: The text introduces the topic of proximal humerus fractures (PHFs) and their prevalence, with emphasis on their occurrence in older patients, often due to falls. The text also highlights complications associated with operative treatments using locking plates.

  2. Rationale for Augmentation: The text discusses the rationale behind augmenting plated complex PHFs with metaphyseal defects, particularly with PMMA-based bone cement, to enhance stability and reduce complications such as screw cut-through and perforation.

  3. Methodology: The study used fresh-frozen human cadaveric humeri with three-part PHFs, and these specimens were divided into two groups – control (no augmentation) and augmented (with PMMA cement augmentation). Biomechanical testing was performed to evaluate stability, construct stiffness, varus deformation, screw migration, screw cut-out, and screw bending.

  4. Results: The study's findings indicate that augmentation with PMMA-based bone cement improved construct stability, reduced varus tilting, and decreased displacement of plated constructs. Additionally, the augmentation reduced the migration, cut-out, and bending of screws, resulting in enhanced fixation stability.

  5. Conclusion: The study concludes that PMMA cement augmentation can improve the stability of plated unstable PHFs with metaphyseal defects, reducing the risk of complications and providing a potentially beneficial technique for clinical practice.

To me no changes are needed apart from checking English by a native.

Author Response

See the Reply to Reviewer 1 in the attached file.

Reviewer 2 Report

The authors studied a new augmentation technique for Metaphyseal Defects of Proximal Humerus Fractures. The research sound interesting however the authors should made some change in the manuscript.

 The author should provide a table of characteristics of different suplermetary materials (autograft, allograft or bone cement …) and explain why they used Polymethyl methacrylate in this experiment.

 The authors should explain how affect the cement viscosity on the fracture reconstruction.

  The Polymethyl methacrylate bone cement has any secondary effect on the human body or in the fracture healing.

 The authors should provide evidence of the bone before and after applied cement.

 In figure 2, the author talk about some rows but the figure does not have any row indication please provide letters in the figures.

 In figure 3 what purpose do they have GT marker, Head maker and shaft Marker?

 at what temperature made the fatigue experiments? because the bones were at -20 °C and the mechanical properties change with temperature.

 There are many references, some are too old and they should be erased

Author Response

See the Reply to Reviewer 2 in the attached file.
